# Sinus Tarsi Morphometry Is Correlated with Flatfoot Severity on Weight-Bearing CT

**DOI:** 10.3390/diagnostics16010162

**Published:** 2026-01-04

**Authors:** Bingshu Chen, Xing Gao, Ying Xu, Tianyuan Zhao, Siyao Yang, Yuan Liu, Bin Jiang, Xihan Zhou, Xiaoqiang Chen, Wencui Li, Jiawei Guo

**Affiliations:** 1Department of Hand and Foot Surgery, Shenzhen Second People’s Hospital (The First Hospital Affiliated to Shenzhen University), Shenzhen 518028, China; bingshu1999@163.com (B.C.); 2310244217@email.szu.edu.cn (X.G.); xuying27713@163.com (Y.X.); liuy09202@gmail.com (Y.L.); ginger_hkd@126.com (B.J.); 13640969608@163.com (X.C.); 2Medical School, Shenzhen University, Shenzhen 518028, China; 3Department of Biomedical Engineering, School of Medicine, Shenzhen University, Shenzhen 518060, China; 15152566232@163.com; 4The Medical Record Department, Shenzhen Second People’s Hospital (The First Hospital Affiliated to Shenzhen University), Shenzhen 518028, China; yang-sy18@tsinghua.org.cn; 5Department of Medicine, SUNY Downstate Health Sciences University, Brooklyn, NY 11203, USA; xihan.zhou@downstate.edu

**Keywords:** ankle joint, pes planus, tarsal bones, orthopedic implants, computed tomography

## Abstract

**Background:** Flexible flatfoot is a common musculoskeletal disorder in adolescents, which is characterized by a collapsed longitudinal arch. A common surgery like subtalar arthroereisis depends on the implant in sinus tarsi. Optimal match between them can potentially avoid postoperative pain and obtain improved prognosis. Investigations into anatomical morphology of sinus tarsi by weight-bearing CT (WBCT) may unveil the pathogenesis and facilitate the treatment of flexible flatfoot. **Methods:** This retrospective study included 28 control cases and 42 flatfoot cases. The sinus tarsi length (STL), the sinus tarsi width (STW), the angle between its long axis and the horizontal line (ST-H angle), the sinus tarsi angle (ST angle), and the tibial width were measured. We also calculated two ratios (STL/tibia width and STW/tibia width) to standardize individual differences. Data analysis was conducted via mean/median comparisons and subsequent linear regression. **Results:** The STL and the STL/tibia width were significantly greater in the flatfoot group (25.73 ± 3.50 vs. 23.09 ± 3.77 mm, *p* = 0.004; 0.90 ± 0.15 vs. 0.81 ± 0.14, *p* = 0.009). The ST angle was significantly smaller in the flatfoot group by an average of 4.63° (13.20° vs. 17.83°, *p* < 0.001). Linear regression revealed that female gender and smaller ST angle were significantly correlated with higher Meary angle, while smaller ST angle and greater STL/tibia width were significantly correlated with lower Pitch angle (*p* = 0.002, *p* = 0.007; *p* = 0.003, *p* = 0.004). No statistical predictive effects were observed for the other variables. **Conclusions:** The ST angle and STL/tibia width may serve as auxiliary parameters for implant selection in subtalar arthroereisis to improve sizing match within the sinus tarsi.

## 1. Introduction

Flatfoot is primarily characterized by the descent of the longitudinal foot arch. Flexible flatfoot is the majority based on foot arch reducibility under weight-bearing (WB) conditions [1]. It is reported that the incidence of flatfoot among children aged 7–14 years is 10.3%, which poses a significant concern to their skeletal health [2]. The collapse of the foot arch leads to altered load distribution in the foot, then causes abnormal alignment of the lower limb joints and even the spine, increasing the risk of joint degeneration and injury. In the long term, this may induce secondary arthritis and synovitis [3,4]. Furthermore, patients suffer pain after prolonged standing or walking, which thereby impairs their quality of life and enthusiasm for physical activities. Conservative treatment is preferred for patients with flexible flatfoot, while surgery is required for cases with poor compliance or severe deformity [5]. Therefore, detailed preoperative morphological measurements of the foot are of great importance for formulating personalized surgical plans.

On WB X-rays, two key indicators for evaluating foot arch collapse are the Meary angle and the Pitch angle [6]. Subtalar joint arthroereisis, the most common procedure, restricts excessive movement of the subtalar joint by inserting an implant into the sinus tarsi [7]. It can effectively correct calcaneal eversion and restore the height of the medial longitudinal arch, thereby improving the Meary angle and Pitch angle. Postoperatively, the stabilized foot arch structure helps restore the patient’s balance and endurance during standing and walking, ultimately alleviating pain and improving gait [8]. Therefore, investigating the morphological differences of hindfoot, particularly the sinus tarsi, is conducive to the formulation of personalized surgical plans [9].

The sinus tarsi is an approximately conical structure, consisting of the medial elongated tarsal canal, the funnel-shaped sinus tarsi, and the tarsal orifice inferior to the tail of the sustentaculum tali [10]. As flatfoot progresses, the space of the sinus tarsi narrows, and the internal pressure increases. This elevated pressure compresses the nerve endings, causing local pain [11]. Without timely intervention, it may evolve to sinus tarsi syndrome, hallux valgus, and metatarsalgia, which severely affect the patient’s quality of life [12]. In addition, some patients experience severe foot pain after subtalar joint arthroereisis, and approximately 40% of them require implant removal [13]. This complication is mainly attributed to the mismatch between the implant and the sinus tarsi [14]. Thus, the surgical outcome partially depends on the preoperative evaluation of the anatomical morphology of the sinus tarsi.

Ma et al. conducted an anatomical study of the sinus tarsi using MRI [11]. Although MRI has advantages in soft tissue evaluation, it is limited to visualize three-dimensional bony structures. Andreas Flury et al. used conventional CT scanning and found that the anatomical structures of the subtalar joint and sinus tarsi in flatfoot patients tend to exhibit plantar flexion and relative medial displacement [15]. However, the non-weight-bearing state introduces errors in measuring joint spaces [16]. Therefore, WBCT is more suitable for evaluating the anatomical morphology of the sinus tarsi and investigating the skeletal deformities with good reliability and reproducibility [17].

In this study, we use WBCT to measure the parameters of the sinus tarsi, including length and width. Through mean/median comparison and linear regression analysis, the anatomical morphological differences of the sinus tarsi under WB conditions between healthy individuals and flatfoot patients are systematically investigated. This study aims to provide a theoretical reference for the radiological diagnosis of flatfoot and the formulation of treatment plans.

## 2. Materials and Methods

### 2.1. Data Collection of the Patients

This single-center retrospective study analyzed WBCT images of patients who underwent ankle WBCT examinations at our institution from August 2023 to July 2025. All WBCT scans were performed using a dedicated WBCT system (Planmed Oy, Helsinki, Finland). The scanning parameters were set as follows: (1) original slice thickness: 0.4 mm, (2) isotropic voxel resolution: 0.4 mm, (3) tube voltage: 96 kVp, and (4) tube current: 10 mA. During the scan, patients assumed a natural single-leg standing position under full WB conditions, ensuring stability and complete plantar load distribution throughout the procedure. The raw imaging data were exported in DICOM format. All DICOM images were imported into Mimics software (version 21.0; Materialise NV, Leuven, Belgium) for annotation of key anatomical landmarks and quantitative measurements, including the length, width, and angle of the sinus tarsi, as well as other evaluated parameters. All image data were sourced from the hospital’s Picture Archiving and Communication System (PACS) and were anonymized prior to analysis by removing identifiable personal information, such as name, contact details, and medical record number.

### 2.2. Radiographic Measurement on WB X-Rays

Three investigators (two board-certified, fellowship-trained musculoskeletal radiologists and one chief orthopedic foot and ankle surgeon with more than 10 years of clinical experience), who were blinded to all clinical information and study design, independently measured the Meary and Pitch angles on WB lateral ankle radiographs using the built-in tools of our hospital’s PACS system. All radiographs were reviewed in random order. For each parameter, measurements from the three readers were averaged, and this mean value was used for subsequent analysis to minimize the effect of any single observer’s measurement variability.

The Meary angle was measured as the angle between the longitudinal axis of the talus and the longitudinal axis of the first metatarsal, with a normal range of 0° ± 4° [11,18]. The Pitch angle was defined as the angle between the inferior border of the calcaneus and the horizontal line, with a normal range of 20–30°. The measurements are exemplified in Figure 1.

The diagnostic criteria for flatfoot in this study were defined as Meary angle > 4° or Pitch angle < 20° [11,18,19].

### 2.3. Inclusion Criteria

(1) Aged between 16 and 65, (2) availability of complete clinical data along with WB X-ray and WBCT images, (3) ability to stand independently and complete the WBCT examination, and (4) willingness to comply with the imaging procedure.

### 2.4. Exclusion Criteria

(1)History of chronic or progressive neuromuscular disorders (e.g., peripheral neuropathy, stroke, Sequelae cerebral palsy, or muscular dystrophy) resulting in significant gait or foot deformities.(2)Previous foot or ankle surgeries (including internal fixation, arthrodesis, or implant placement) or the presence of unhealed fractures within the past one year.(3)Diagnosed inflammatory joint diseases (e.g., rheumatoid arthritis or ankylosing spondylitis) or systemic metabolic bone diseases (e.g., severe osteoporosis with a history of fractures, or osteomalacia).(4)Congenital foot deformities (e.g., congenital vertical talus or polydactyly/syndactyly with malformation) or other significant structural congenital abnormalities.(5)Regular use of custom orthotic insoles or arch supports for corrective purposes (e.g., plantar fasciitis or Charcot foot).(6)Prior systematic physical therapy or rehabilitation training targeting the foot and ankle (e.g., ankle sprain or achilles tendon rupture).

### 2.5. Morphometric Measurements of the Sinus Tarsi on WBCT

We analyzed the WBCT images by using the Mimics imaging software. The anterior subtalar joint is separated from the posterior subtalar joint by the tarsal canal and sinus tarsi [20]. Therefore, in the coronal plane, images were scrolled from posterior to anterior, and the region of interest was defined as the area at the anterior margin of the anterior subtalar joint where the sustentaculum tali of the calcaneus was about to completely disappear. Within three consecutive slices in this region, the slice with the largest cross-sectional area of the sinus tarsi was selected for measurement (Figure 2).

The following parameters were measured: sinus tarsi length (STL), sinus tarsi width (STW), the angle between the long axis of the sinus tarsi and the horizontal line (ST-H angle), the sinus tarsi angle (ST angle), and tibia width. The ratios of STL to tibia width (STL/tibia width) and STW to tibia width (STW/tibia width) were subsequently calculated. The measured parameters were defined as the following and are marked in Figure 3:(1)Sinus tarsi length (STL): the distance between the medial apex of the sinus tarsi canal (at the talar edge of the anterior subtalar articular facet) and the lateral endpoint of the sinus tarsi canal orifice on the calcaneus.(2)Sinus tarsi width (STW): the vertical distance between the superior and inferior borders of the sinus tarsi canal, measured along a line perpendicular to the talar surface of the tibiotalar joint, passing through the midpoint of that surface.(3)Angle between the long axis of the sinus tarsi and the horizontal line (ST-H angle): the angle formed by the line connecting the medial apex of the sinus tarsi canal (at the talar edge of the anterior subtalar articular facet) to the lateral endpoint of the sinus tarsi canal orifice on the calcaneus, and the horizontal line.(4)Sinus tarsi angle (ST angle): the angle formed by the lateral superior point and lateral inferior point of the sinus tarsi orifice, and the medial apex of the sinus tarsi canal (the talar edge of the anterior subtalar articular facet).(5)Tibia width: the distance between the lateral edge of the tibial plafond and the medial turning point of the tibia at the tibiotalar joint from an anterior–posterior view.

The same three investigators, blinded to clinical information and study design, independently measured each CT-derived parameter. All CT images were assessed in random order, and the mean of the three measurements for each parameter was recorded and used for analysis. Interobserver and intraobserver reliability for both radiographic and CT measurements were assessed using the intraclass correlation coefficient (ICC) with corresponding 95% confidence interval (95% CI). For intraobserver reliability, all three observers repeated all X-ray and CT measurements in a new random order at least 3 weeks after the initial assessment (Appendix A).

For interobserver reliability, all parameters demonstrated substantial to almost perfect agreement, with ICC values ranging from 0.793 to 0.886. For intraobserver reliability, the ICC values for all parameters across the three observers ranged from approximately 0.882 to 0.970, also indicating almost perfect agreement and high measurement reliability.

### 2.6. Statistical Analysis

All statistical analyses were performed using SPSS statistical software (version 24.0; IBM Corp., Armonk, NY, USA). Continuous data were first assessed for normality by the Shapiro–Wilk test. Data with *p* > 0.05 were considered normally distributed and presented as mean ± SD. Data with *p* < 0.05 were considered non-normally distributed and presented as median (Q1, Q3). Categorical variables were summarized as counts (percentages). Based on data type and distribution, appropriate statistical tests were selected. Categorical variables were compared using Yates’s corrected chi-square test or Pearson’s chi-square test. Normally distributed continuous variables were compared between groups using the independent samples *t* test. If variances were unequal, Welch’s *t* test was applied. Non-normally distributed continuous variables were compared using the Mann–Whitney U test. Two-tailed tests were used, with *p* < 0.05 considered statistically significant.

A stepwise multiple linear regression analysis was performed. Significance thresholds for variable inclusion (*p* ≤ 0.05) and exclusion (*p* ≥ 0.10) were established, followed by sequential screening of candidate independent variables. A regression model was finally derived that only included independent variables with a significant independent contribution to the dependent variable and no severe multicollinearity (tolerance < 0.1). Additionally, analysis of variance (ANOVA) was used to evaluate the overall validity of the model, while adjusted R^2^ was employed to assess the model’s explanatory power for variations in the dependent variable.

## 3. Results

According to the predefined inclusion and exclusion criteria, 70 patients were included in this study for the subsequent analysis (Figure 4). Table 1 summarizes the demographic and baseline clinical characteristics of the participants, including age, body mass index, and radiographic measurements (the Meary angle and Pitch angle).

As detailed in Table 2, the 70 participants were categorized into a control group (*n* = 28) and flatfoot group (*n* = 42) based on the diagnostic criteria for flatfoot. Detailed intragroup comparison of demographic and clinical characteristics was conducted. A significant difference was observed in the sex distribution between the groups (*p* = 0.015). Additionally, participants in the control group were significantly taller than those in the flatfoot group (*p* = 0.019). However, no statistically significant differences were found in age, weight, body mass index, or the affected side. The control group exhibited a significantly lower median Meary angle (3° vs. 10°, *p* < 0.001) and a significantly larger median Pitch angle (24.5° vs. 16°, *p* < 0.001) compared to the flatfoot group.

As shown in Table 3, multiple morphological parameters of the sinus tarsi were compared to further investigate the impact of flatfoot. Average STL was significantly higher in the flatfoot group (25.73 mm vs. 23.09 mm, *p* = 0.004). Conversely, median ST angle was significantly lower in the flatfoot group (13.20° vs. 17.83°, *p* < 0.001). To account for the influence of individual body size, the average STL/tibia width was calculated. It was also significantly larger in the flatfoot group (0.90 vs. 0.81, *p* = 0.009). No statistically significant differences were observed in the comparisons of STW, ST-H angle, tibial width, or STW/tibia width.

To identify the most critical sinus tarsi parameters influencing arch angles, stepwise linear regression analyses were conducted. We first regarded intercept and ST angle as predictive variables and evaluated the predictive effects of gender, height, STL, and STL/tibia width by assuming excluded variables.

As shown in Appendix A, when the Meary angle was the dependent variable, Model 1 (predictive variable: ST angle; excluded predictive variables: gender, height, STL, and STL/tibia width) demonstrated significant predictive effects (*p* < 0.001). Then, STL/tibia width was added in the following Model 2 (predictive variables: ST angle and STL/tibia width; excluded predictive variables: gender, height, and STL) and showed statistical significance (*p* = 0.003, *p* = 0.010). Then, gender was subsequently added in Model 3 (predictive variables: gender, ST angle, and STL/tibia width; excluded predictive variables: height and STL) and remained significant (*p* = 0.002, *p* = 0.007, *p* = 0.076). Meanwhile, the assumed excluded variables were insufficient to be included in the next model (*p* > 0.05). Ultimately, gender, ST angle, and STL/tibia width were retained as the core predictors in the final equation.

As shown in Appendix A, when the Pitch angle was the dependent variable, Model 1 (predictive variable: ST angle; excluded predictive variables: gender, height, STL, and STL/tibia width) demonstrated significant predictive effects (*p* < 0.001). Then, STL/tibia width was added in the following Model 2 (predictive variables: ST angle and STL/tibia width; excluded predictive variables: gender, height, and STL) and showed statistical significance (*p* = 0.003, *p* = 0.004). Meanwhile, the assumed excluded variables were insufficient to be included in the next model (*p* > 0.05). Ultimately, ST angle and STL/tibia width were retained as the core predictors in the final equation.

Collectively, as shown in Table 4, when the Meary angle was set as the dependent variable, gender, ST angle, and STL/tibia width were identified as significant predictors, respectively (B = −5.098, t = −3.254, *p* = 0.002; B = −0.180, t = −2.789, *p* = 0.007; B = 8.825, t = 1.803, *p* = 0.076). Besides, gender demonstrated the greatest predictive effect because of the highest |β| (0.344), followed by ST angle (0.289) and STL/tibia width (0.193).

When the Pitch angle was set as the dependent variable, ST angle and STL/tibia width were identified as significant positive and negative predictors, respectively (B = 0.197, t = 3.079, *p* = 0.003; B = −14.134, t = −3.008, *p* = 0.004). Besides, ST angle demonstrated greater predictive effect than STL/tibia width because of a higher |β| (0.332 vs. 0.325).

Subsequently, regression ANOVA was conducted to evaluate the overall validity of each stepwise model (Appendix A). The results demonstrated that two final models predicting the Meary angle or Pitch angle were both statistically significant (F = 12.112, *p* < 0.001; F = 12.559, *p* < 0.001; Table 5). Furthermore, we evaluated their predictive performance and explanatory power. The final model predicting the Meary angle explained 32.6% of the variance, while the model predicting the Pitch angle explained 25.1% of the variance (Appendix A).

## 4. Discussion

The sinus tarsi is a frequently used surgical approach and operational channel for subtalar joint arthroereisis [21,22]. However, previous studies were unable to accurately reflect its true anatomical morphology under limb WB conditions. To achieve better matching between implants and the sinus tarsi and improve patient prognosis, this study used WBCT from a three-dimensional perspective to identify significant differences in height, gender, STL, ST angle, and STL/tibia width between healthy individuals and flatfoot patients.

Our results revealed a significantly greater STL in the flatfoot group compared to the control group. This finding indicates structural differences in the sinus tarsi of flatfoot patients. To minimize the influence of individual body size, we standardized STL/tibia width, considering that previous studies have predominantly focused on STL alone. Subsequent linear regression analysis identified STL/tibia width as more strongly correlated with Meary and Pitch angles than STL alone, suggesting it is a better predictor for flatfoot evaluation and preoperative planning.

These findings gain clinical relevance in light of existing literature. Postoperative sinus tarsi pain is a common complication following subtalar joint arthroereisis, with potential causes including overcorrection, undercorrection, implant impingement, or soft tissue irritation [23]. Moreover, Cook et al. demonstrated in a propensity score-matched study that inadequate correction of the Meary angle increases the risk of implant removal by 17.5% (*p* = 0.0012) [24]. Therefore, utilizing ST angle and STL/tibia width as standardized preoperative parameters for implant selection may not only enhance the precision of arch correction but also help reduce the occurrence of postoperative sinus tarsi pain.

The results in this study identified female gender as an independent predictor of an increased Meary angle, indicating a higher susceptibility to flatfoot in females. We hypothesize that this observation is attributed to relatively weaker muscular strength and greater ligamentous laxity in females, which collectively impair the structural integrity of the foot arch [25]. Consistent with this, Diwakar et al. have also documented a higher prevalence of flatfoot in females compared to males (13.95% vs. 2.27%) [26]. Notably, the control group exhibited significantly greater height than the flatfoot group. For one thing, this discrepancy may be partially confounded by the imbalanced sex ratio between the two groups (24/4 vs. 23/19), rather than reflecting a true biological association. For another, in linear regression analyses adjusting for height as a covariate, it was not included in the final models for predicting Meary or Pitch angles. This indicates that height does not exert a statistically significant independent predictive effect on these angular parameters (Appendix A). It aligns with previous epidemiological evidence demonstrating no significant correlation between height and the prevalence of flatfoot.

We found a significant reduction in the ST angle among flatfoot patients compared to the control group, indicating notable narrowing of the sinus tarsi orifice. Linear regression analysis further verified that the ST angle was negatively correlated with the Meary angle (B = −0.180, t = −2.789, *p* = 0.007) and positively correlated with the Pitch angle (B = 0.197, t = 3.079, *p* = 0.003). These findings suggest that the ST angle has predictive value for assessing arch structural parameters and reinforce the close relationship between sinus tarsi morphology and the structural alterations associated with flatfoot, which is consistent with and extends previous morphological studies. Using MRI, Yong et al. reported significantly increased sinus tarsi length and width in flatfoot patients (*p* < 0.001) and proposed that disease progression may involve sinus tarsi narrowing, leading to elevated compartmental pressure and nerve compression that underlies pain [27]. Clinical observations further support the high prevalence of sinus tarsi pain in this population [13,27]. Our angular findings align with morphological descriptions by Cody et al., who used WBCT and identified significantly increased talar eversion in flatfoot patients (*p* < 0.01) [28]. Taken together, subtalar joint excessive internal rotation secondary to medial longitudinal arch collapse [29] and increased talar eversion may act synergistically to narrow the sinus tarsi orifice, thereby contributing to the progression of flatfoot deformity. However, our linear regression models incorporating ST angle and STL/tibia width explained only a small proportion of the variance in Meary angle and Pitch angle despite their statistical significance (adjusted R^2^ = 32.6% and adjusted R^2^ = 25.1%). This suggests that alterations in foot alignment are modulated by the complex interplay of multiple anatomical and biomechanical factors, with sinus tarsal morphology measured in our study representing merely one such variable.

## 5. Limitation

Most studies focused on the talar morphology, while flatfoot development involves the alignment changes of the entire foot skeletal structure [4,28,30,31]. While our study supplements the existing understanding of this issue, certain limitations must be recognized. First, the relatively small sample size and the observed baseline disparities, such as in sex and height, may have influenced the outcomes. Furthermore, potential confounding factors, including race and anatomical variations, were not accounted for in this study. Second, the single-center, retrospective design is susceptible to selection bias, while the reliance on two-dimensional coronal views may have introduced measurement variability in the assessment of three-dimensional structures. Third, the use of retrospective imaging data with incompletely standardized positioning could have affected the identification and measurement of the sinus tarsi morphology. In addition, the static WB analysis did not encompass dynamic functional assessments, thereby limiting our understanding of the sinus tarsi configuration during physiological motion. Finally, the lack of postoperative follow-up and the inability to directly correlate sinus tarsi morphology with implant outcomes preclude any evaluation of long-term prognosis.

## 6. Conclusions

A lower ST angle and higher STL/tibia width were positively correlated with flatfoot severity evaluated by Meary or Pitch angles. Additionally, female gender was positively correlated with flatfoot severity diagnosed by Meary angle.

## Figures and Tables

**Figure 1 diagnostics-16-00162-f001:**
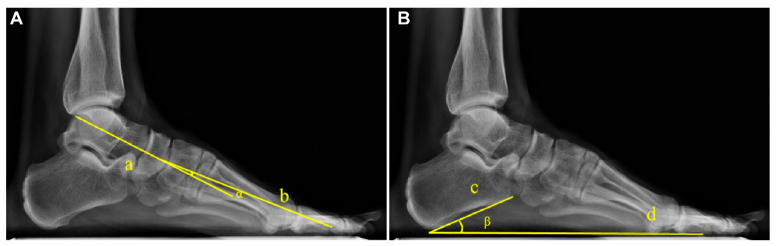
Example diagram of arch angle measurement based on WB X-rays. (**A**) The Meary angle (α): formed by the longitudinal axis of the talus (Line a) and the longitudinal axis of the first metatarsal (Line b). (**B**) The Pitch angle (β): formed between the tangent to the inferior border of the calcaneus (Line c) and the line connecting the lowest point of the calcaneus to the lowest point of the medial sesamoid of the forefoot (Line d).

**Figure 2 diagnostics-16-00162-f002:**
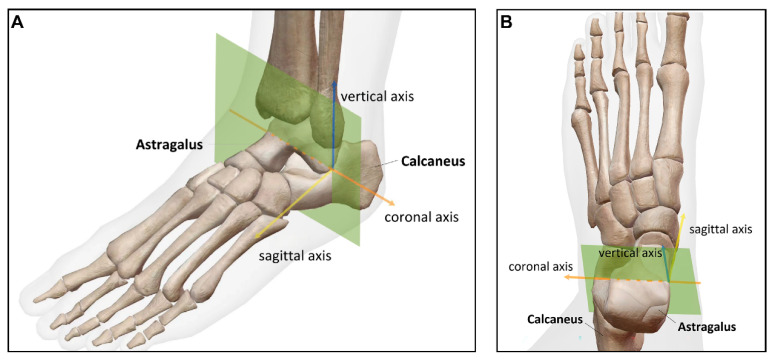
Schematic diagram of image measurement section based on WBCT. The selected measurement section (green section) demonstrated from the obliquely upper view (**A**) and the top-down view (**B**). Image resource: Visible Body Suite (Version 6.3.12) (Computer software). (2025). Retrieved 10 October 2025 from https://www.visiblebody.com/.

**Figure 3 diagnostics-16-00162-f003:**

Example diagram of the analyzed parameters from WBCT. (**A**) Sinus tarsi length (STL): the length between the medial apex (Point a) of the sinus tarsi canal (at the talar edge of the anterior subtalar articular facet) and the inferolateral endpoint (Point b) of the sinus tarsi canal orifice. (**B**) Sinus tarsi width (STW): the vertical width of the sinus tarsi canal (the length of Line c–d), measured along a perpendicular line passing through the midpoint of Line e–f on the talar surface of the tibiotalar joint, where Point c and Point d denote the intersections of this perpendicular with the superior and inferior borders of the sinus tarsi canal, respectively. (**C**) The angle between the long axis of the sinus tarsi and the horizontal line (ST–H angle) (β): formed by the long axis of the sinus tarsi canal (Line a–b) and the horizonal line. (**D**) Sinus tarsi angle (ST angle) (α): formed by the superolateral point and inferolateral point of the sinus tarsi orifice and the medial apex (Point a) of the sinus tarsi canal (i.e., the talar edge of the anterior subtalar articular facet). (**E**) Tibia width: the length of Line g–h connected the lateral edge (Point g) and the medial turning point (Point h) of the tibial plafond.

**Figure 4 diagnostics-16-00162-f004:**
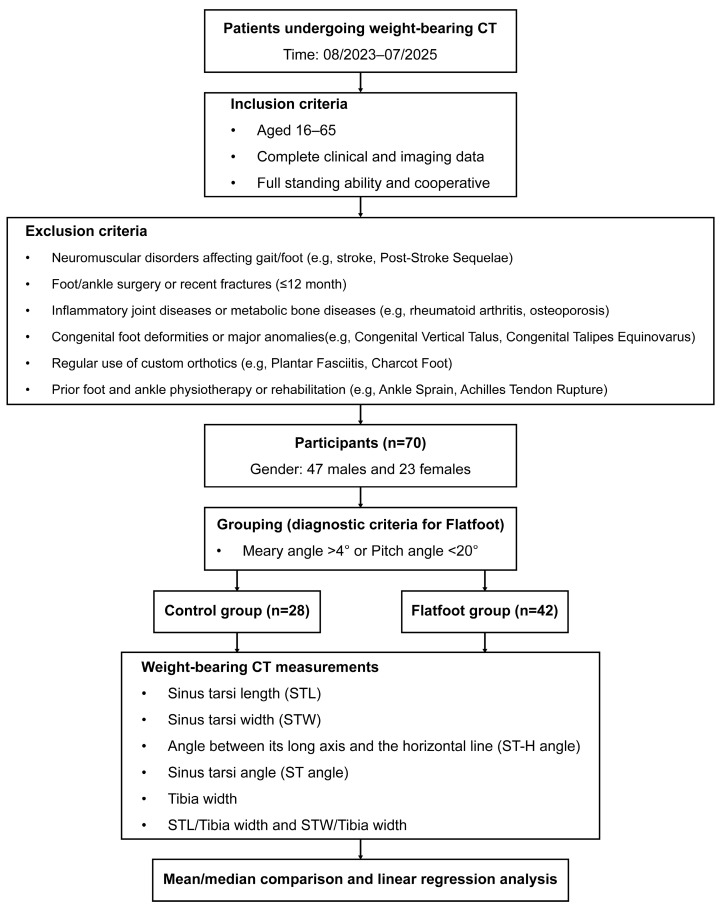
Flowchart of the enrollment, grouping, and analysis.

**Table 1 diagnostics-16-00162-t001:** Demographic and clinical data of the 70 participants.

	Min.	Max.	Mean ± SD	Median (Q1, Q3)
Age (Y)	18	62	36.36 ± 10.59	36 (27, 43)
Height (m)	1.53	1.88	1.69 ± 0.07	1.70 (1.63, 1.73)
Weight (kg)	43.4	100.0	70.15 ± 11.10	70.5 (62.0, 78.0)
BMI (kg/m^2^)	18.06	34.60	24.49 ± 3.08	24.86 (22.05, 26.13)
Meary angle (°)	−4	33	8.03 ± 7.03	6 (3.75, 6)
Pitch angle (°)	2	29	18.93 ± 6.66	20 (15, 24.25)

BMI: body mass index.

**Table 2 diagnostics-16-00162-t002:** Demographic and clinical data stratified by the Meary angle or Pitch angle.

	Control (*n* = 28)	Flatfoot (*n* = 42)	Statistics	*p*-Value
Gender (M/F)	24/4	23/19	5.96	**0.015** ^a^
Side (L/R)	14/14	22/20	0.038	0.845 ^b^
Age (Y)	37.39 ± 8.77	35.67 ± 11.70	0.666	0.508 ^c^
Height (m)	1.71 ± 0.06	1.67 ± 0.08	2.405	**0.019** ^d^
Weight (kg)	72.88 ± 11.59	68.32 ± 10.51	1.705	0.093 ^c^
BMI (kg/m^2^)	24.83 ± 3.52	24.27 ± 2.77	0.742	0.460 ^c^
Meary angle (°)	3 (1, 4)	10 (7, 14)	−7.008	**<0.001** ^e^
Pitch angle (°)	24.5 (21, 27)	16 (11.75, 19)	−5.879	**<0.001** ^e^

M: male, F: female, L: left, R: right, BMI: body mass index. Data were presented as mean ± SD or median (Q1, Q3). a: Yates’s chi-squared test, b: Pearson’s chi-squared test, c: non-paired Student’s *t* test, d: Welch’s *t* test, e: Mann–Whitney U test. Bolded *p*-values indicate statistical significance at *p* < 0.05.

**Table 3 diagnostics-16-00162-t003:** The comparisons of the parameters in sinus tarsi between control and flatfoot groups.

	Control (*n* = 28)	Flatfoot (*n* = 42)	Statistics	*p*-Value
STL (mm)	23.09 ± 3.77	25.73 ± 3.50	−3.000	**0.004 ^a^**
STW (mm)	8.71 ± 1.25	8.33 ± 2.07	0.937	0.352 ^b^
ST-H angle (°)	3.93 ± 4.15	2.89 ± 7.81	0.725	0.471 ^b^
ST angle (°)	17.83 (16.06, 23.24)	13.20 (6.29, 18.40)	−3.501	**<0.001 ^c^**
Tibia width (mm)	28.92 ± 3.59	28.87 ± 3.42	0.058	0.954 ^a^
STL/tibia width	0.81 ± 0.14	0.90 ± 0.15	−2.680	**0.009 ^a^**
STW/tibia width	0.31 ± 0.06	0.29 ± 0.07	0.958	0.341 ^a^

STL: sinus tarsi length, STW: sinus tarsi width, ST: sinus tarsi, H: horizontal line. Data were presented as mean ± SD or median (Q1, Q3). a: Non-paired Student’s *t* test, b: Welch’s *t* test, c: Mann–Whitney U test. Bolded *p*-values indicate statistical significance at *p* < 0.05.

**Table 4 diagnostics-16-00162-t004:** Linear regression equation.

**Meary Angle as Dependent Variable.**					
	**Unstandardized**		**Standardized**		
	**B**	**SER**	**β**	**t**	** *p* ** **-value**
Intercept	6.478	4.916		1.318	0.192
Gender *	−5.098	1.567	−0.344	−3.254	**0.002**
ST angle (°)	−0.180	0.065	−0.289	−2.789	**0.007**
STL/tibia width	8.825	4.894	0.193	1.803	0.076
**Pitch Angle as Dependent Variable.**					
	**Unstandardized**		**Standardized**		
	**B**	**SER**	**β**	**t**	** *p* ** **-value**
Intercept	28.270	4.446		6.358	**<0.001**
ST angle (°)	0.197	0.064	0.332	3.079	**0.003**
STL/tibia width	−14.134	4.699	−0.325	−3.008	**0.004**

ST: sinus tarsi, STL: sinus tarsi length, SER: standard error of the regression, β: standardized regression coefficient. *: Female = 0, male = 1. Bolded *p*-values indicate statistical significance at *p* < 0.05.

**Table 5 diagnostics-16-00162-t005:** Regression ANOVA.

**Meary Angle as Dependent Variable.**					
	**Sum of Square**	**df**	**Mean Square**	**F**	** *p* ** **-value**
Regression	1204.994	3	401.665	12.112	**<0.001**
Residual	2188.778	66	33.163		
Total	3393.771	69			
**Pitch Angle as Dependent Variable.**					
	**Sum of Square**	**df**	**Mean Square**	**F**	** *p* ** **-value**
Regression	835.110	2	417.555	12.559	**<0.001**
Residual	2227.533	67	33.247		
Total	3062.643	69			

Bolded *p*-values indicate that the overall regression model is statistically significant at *p* < 0.05.

## Data Availability

The data that support the findings of this study are available from the corresponding author upon reasonable request.

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
