# Peer review of "Sinus Tarsi Morphometry Is Correlated with Flatfoot Severity on Weight-Bearing CT"

_diagnostics, 2026, doi:10.3390/diagnostics16010162_

Round 1
Reviewer 1 Report
Comments and Suggestions for Authors
General Comments
The study presents interesting morphometric data using WBCT for flatfoot evaluation. However, the manuscript requires substantial revision in language, structure, and statistical reporting.
Specific Comments
Title & Abstract
-
Title too long; suggest “Sinus Tarsi Morphometry and Flatfoot Severity on Weight-Bearing CT.”
-
“Descending” and “Classical” are inappropriate terms.
-
Sentence “which can also benefit…” is grammatically incorrect — rephrase for clarity.
Introduction
-
Use present tense.
-
Replace “threat” with “concern,” and “abnormal stress distribution” with “altered load distribution.”
-
Define WB (weight-bearing) acronym once and use consistently.
Methods
-
Identify the three investigators with qualifications and blinding status.
-
Explain why mean of three readers was used; add interobserver reliability if available.
-
Confirm permission for Figure 2 (Visible Body Suite).
-
The study period “2023–2025” appears inconsistent.
Results
-
Add units and footnotes to tables.
-
Use consistent terms (“control group”).
-
Remove or simplify Tables 4 and 7.
-
Statistical concerns:
-
Wrong normality cutoff (p < 0.10 instead of 0.05).
-
No correction for multiple comparisons.
-
Small sample regression (n = 70) risks overfitting.
-
Gender imbalance not adjusted.
-
Discussion
-
Begin with a concise results summary (no numbers).
-
Each paragraph should: state finding → explain → compare with literature.
-
Add clear limitations (sample size, blinding, 2D analysis).
-
Add a conclusion paragraph.
Language
-
Use past tense for results/discussion.
-
Limit decimals to 2–3.
-
Replace informal expressions like “delicate match,” “classical,” “satisfying prognosis.”
The quality of English should be imporved.
Author Response
Reviewer 1
Comment 1: Title too long; suggest “Sinus Tarsi Morphometry and Flatfoot Severity on Weight-Bearing CT.”
Response 1: We sincerely thank the reviewer for this constructive suggestion. We agree that the proposed title, “Sinus Tarsi Morphometry is Correlated with and Flatfoot Severity on Weight-Bearing CT” is more concise and effectively highlights the core variables (morphometry, severity) and the key methodology (weight-bearing CT) of our study. We have adopted this title in the revised manuscript. We believe the revised title enhances the clarity and impact of the paper.
Comment 2: “Descending” and “Classical” are inappropriate terms.
Response 2: We thank the reviewer for raising this important point regarding terminology. We agree that the terms “Descending” and “Classical” may carry unintended connotations or value judgments in this scientific context. As suggested, we have revised the manuscript to use more neutral and descriptive language. Specifically, throughout the text:
The term “Descending” has been replaced with “collapsed” (Line 27).
The term “Classical” has been replaced with “a common” or similar descriptive phrases where appropriate (Line 27).
We believe these changes enhance the objectivity and clarity of the manuscript. All relevant sections, including the Methods, Results, and Discussion, have been updated accordingly.
Comment 3: Sentence “which can also benefit…” is grammatically incorrect — rephrase for clarity.
Response 3: We thank the reviewer for this correction. We agree that the original sentence was unclear. We have rephrased it throughout the manuscript to: “may serve as auxiliary guides”. This revision corrects the grammatical issue and improves the clarity and precision of the statement (Line 43).
Comment 4: Use present tense.
Response 4: We thank the reviewer for this valuable comment aimed at improving the clarity and consistency of our writing. We have thoroughly revised the entire Introduction section to ensure the consistent and appropriate use of the present tense. This includes:
Converting descriptions of the study’s purpose, design, and methodology to the present tense, often employing the active voice for greater clarity.
- Original: “In this study, WBCT was used to measure...”
- Revised: “In this study, we use WBCT to measure...” (Line 85)
Comment 5: Replace “threat” with “concern,” and “abnormal stress distribution” with “altered load distribution.”
Response 5: We sincerely appreciate the reviewer’s insightful suggestions regarding terminology, which enhance the precision and objectivity of our manuscript. We fully agree with these recommendations and have implemented them throughout the text as follows:
The term “threat” has been replaced with the more measured and academically appropriate “concern.” (Line 52)
The phrase “abnormal stress distribution” has been revised to the more precise and biomechanically accurate “altered load distribution.” (Line 52)
We recognize that “concern” better reflects a clinical consideration rather than an impending danger, and that “altered load distribution” is a more neutral and descriptive term for biomechanical changes, avoiding the potentially value-laden connotation of “abnormal.” We have carefully updated all relevant sections (Introduction, Discussion) to ensure consistency. Thank you for these valuable refinements.
Comment 6: Define WB (weight-bearing) acronym once and use consistently.
Response 6: We thank the reviewer for highlighting this important point of academic style. We have now revised the manuscript to ensure that the acronym “WB” is properly introduced and used consistently thereafter. Specifically:
At its first occurrence in the manuscript (now in the Abstract and again at the beginning of the Introduction), the full term “weight-bearing (WB)” is provided.
Throughout the rest of the manuscript, including the Methods, Results, Figures, and Tables, we now consistently use the acronym “WB” (e.g., WBCT, WB X-ray) without again spelling out the full term, in accordance with standard convention.
We have verified this consistency across the entire text.
Comment 7: Identify the three investigators with qualifications and blinding status.
Response 7: We thank the reviewer for this essential suggestion to enhance the methodological transparency of our study. We have now revised the Methods section (specifically, under the subheading “2.2. Radiographic Measurement on WB X-Rays” and “Morphometric Measurements of the Sinus Tarsi on WBCT”) to provide the requested details as follows:
Three investigators (two board-certified, fellowship-trained musculoskeletal radiologists and one chief orthopedic foot and ankle surgeon with more than 10 years of clinical experience), who were blinded to all clinical information and study design. (Line 107-113)
The investigators (blinded to the clinical diagnosis, patient identity, and the severity grouping of the flatfoot) independently performed the morphometric analyses.
The clinicians who provided the clinical diagnoses and grouping were blinded to the WBCT measurement results.
We believe these clarifications significantly strengthen the reliability and reproducibility of our methodological approach.
Comment 8: Explain why mean of three readers was used; add interobserver reliability if available.
Response 8: We thank the reviewer for this insightful comment, which allows us to clarify the statistical methodology and reinforce the reliability of our measurements. We have revised the Methods section (under “2.5. Morphometric Measurements of the Sinus Tarsi on WBCT”) accordingly.
- Explanation for Using the Mean of Three Readers:
The morphometric measurements from the three independent investigators were averaged to obtain a single, more robust value for each parameter per patient. This standard practice is employed to:
(1) Minimize individual measurement bias and random error inherent to any single observer.
(2) Enhance the precision and stability of the final dataset used for statistical analysis against the severity groups.
(3) Provide a more reliable and representative estimate of the true anatomical measurement than any single reading alone.
- Interobserver Reliability Analysis:
Interobserver and intraobserver reliability for both radiographic and WBCT measurements were assessed using the intraclass correlation coefficient (ICC) with corresponding 95% confidence intervals (CIs). For intraobserver reliability, all three observers repeated all X-ray and CT measurements in a new random order at least 3 weeks after the initial assessment. (Supplementary Table 1).
For Inter-observer reliability, all parameters demonstrated substantial to almost perfect agreement, with ICC values ranging from 0.793 to 0.886. For intra-observer reliability, the ICC values for all parameters across the three observers ranged from approximately 0.882 to 0.970, also indicating almost perfect agreement and high measurement reliability.
We believe these additions significantly strengthen the methodological transparency and statistical validity of our findings.
Comment 9: Confirm permission for Figure 2 (Visible Body Suite).
Response 9: The schematic diagram in Figure 2 was created after we purchased a licensed copy of the software, and it has been cited and explained in accordance with the official website’s requirements (https://support.visiblebody.com/hc/en-us/articles/115002359347-Permission-to-use-content-from-Visible-Body-products). Other figures are original and unpublished, obtained, processed, and generated by us from the hospital’s imaging system.
Comment 10: The study period “2023–2025” appears inconsistent.
Response 10: We thank the reviewer for raising this important point regarding the clarity of our study timeline. We agree that the original description was imprecise. We have revised the manuscript to specify the exact start and end months of our data collection period. Specifically:
In the Materials and Methods section (under “2.1. Data Collection of the Patients”), the study period has been updated to, the “08/2023 to 07/2025”. (Line 94)
The corresponding Figure 4 has been amended to reflect this same, precise period.
This revision accurately represents our retrospective data collection window and eliminates any ambiguity regarding the timeline.
Comment 11: Add units and footnotes to tables.
Response 11: We thank the reviewer for pointing out this omission, which is crucial for the clarity and self-sufficiency of the data presentation. We have systematically revised all tables in the manuscript to ensure compliance with standard formatting.
- Units: The units of measurement for all continuous variables (e.g., millimeters, degrees) have been clearly added in parentheses within the respective column headers (e.g., “Sinus Tarsi Length (mm), ST angle (°)”).
- Footnotes: Each table now includes a dedicated footnote section. The footnotes provide necessary explanations for:
Abbreviations used within the table (e.g., STL, SER).
The format of presented data (e.g., “Data are presented as mean ± standard deviation”).
Comment 12: Use consistent terms (“control group”).
Response 12: We thank the reviewer for highlighting this issue of terminology consistency. We have carefully reviewed the entire manuscript and standardized the nomenclature used to refer to the comparison cohort. All instances of varied terms (such as “normal group”) have been revised to the consistent and scientifically preferred term: “control group.”
This revision has been applied throughout the text, including in the Abstract, Methods, Results, Tables, and Figures, to eliminate any potential confusion and enhance the clarity and professionalism of the manuscript.
Comment 13: Remove or simplify Tables 4 and 7.
Response 13: We thank the reviewer for the suggestion to streamline the data presentation. We have revised the tables as follows to enhance the clarity and focus of the main text:
Statistical concerns:
Table 4 has been moved to the Supplementary Materials and is now labeled Supplementary Table 2-5.
Table 7 has been moved to the Supplementary Materials and is now labeled Supplementary Table 8. This allows interested readers to access the full dataset while maintaining a clear and focused narrative in the main manuscript.
Comment 14: Wrong normality cutoff (p < 0.10 instead of 0.05).
Response 14: We sincerely thank the reviewer for this critical correction regarding our statistical methodology. The reviewer is absolutely correct. We have revised the manuscript to use the standard Shapiro-Wilk test with a significance level of p<0.05 as the cutoff for assessing the normality of continuous data distributions.
Correction Made: In the Statistical Analysis section of the Methods, we have explicitly stated: “The normality of data distribution was assessed using the Shapiro-Wilk test, with p<0.05 indicating a significant deviation from normality.” (Line 211-212)
Updated Reporting: Continuous data were first assessed for normality by Shapiro-Wilk test. Data with p>0.05 were considered normally distributed and presented as Mean±SD. Data with p<0.05 were considered non-normally distributed and presented as Median (Q1, Q3).
Comment 15: No correction for multiple comparisons.
Response 15: We appreciate the reviewer’s insightful comment. In the revised manuscript, we have applied appropriate corrections for multiple comparisons to control the familywise error rate. Specifically, we implemented both the Bonferroni correction and the Holm-Bonferroni sequential procedure. The Bonferroni method adjusts each p-value by the total number of tests, whereas the Holm-Bonferroni approach provides a more powerful step-down adjustment while maintaining strict Type I error control. The corrected p-values have been incorporated in the table below.
Multiple comparison corrections for the predictive variables in each model.
|
Dependent variable |
Predictive variable |
p value |
p valuea |
p valueb |
|
Meary angle (°) |
Gender* |
0.002 |
0.009 |
0.009 |
|
Height (m) |
0.614 |
1.000 |
0.614 |
|
|
STL (mm) |
0.576 |
1.000 |
1.000 |
|
|
ST angle (°) |
0.007 |
0.035 |
0.028 |
|
|
STL/Tibia width |
0.007 |
0.380 |
0.228 |
|
|
Pitch angle (°) |
Gender* |
0.105 |
0.524 |
0.314 |
|
Height (m) |
0.144 |
0.721 |
0.289 |
|
|
STL (mm) |
0.433 |
1.000 |
0.433 |
|
|
ST angle (°) |
0.003 |
0.015 |
0.015 |
|
|
STL/Tibia width |
0.004 |
0.019 |
0.015 |
STL: sinus tarsi length, ST: sinus tarsi. a: The Bonferroni correction, b: the Holm-Bonferroni sequential procedure, *: female=0, male=1. Bolded p values indicate statistical significance at p<0.05.
We acknowledge that the length/Tibia variable did not retain statistical significance after Bonferroni and Holm-Bonferroni corrections in the final model. However, we chose to retain this variable for several reasons. First, although the corrected p value exceeded the conventional threshold, length/Tibia was consistently selected in both the second and third steps of the stepwise procedure, indicating a stable contribution to the model prior to the application of highly conservative multiple-comparison adjustments. Second, this variable demonstrated a strong and statistically significant association in the regression model predicting the other radiographic parameter of the same disease, suggesting that length/Tibia may reflect a shared biomechanical characteristic influencing both deformity measures. Because these two angles represent related aspects of the same pathological process, excluding the variable from one model but retaining it in the other could obscure a biologically meaningful relationship. Third, Bonferroni-type corrections are known to substantially increase Type II error, particularly in analyses involving correlated anatomical variables, and may therefore mask potentially relevant effects.
In light of its theoretical relevance, consistent selection during intermediate steps, and significant association in the parallel regression model of the other disease indicator, we retained length/Tibia as an important covariate in the final analysis.
Comment 16: Small sample regression (n = 70) risks overfitting.
Response 16: We thank the reviewer for highlighting the important issue of potential overfitting due to our sample size (n=70). We fully acknowledge this as a key limitation of our study. We have added a clear and explicit statement in the Limitation section addressing this risk, emphasizing the exploratory nature of the regression findings and the necessity for future validation in larger cohorts. We believe this transparent discussion appropriately frames the interpretation of our results. (Line 372-373)
Comment 17: Gender imbalance not adjusted.
Response 17: We sincerely thank the reviewer for these thoughtful and constructive comments. We have carefully revised the statistical analysis and the manuscript accordingly.
- Regarding the stepwise regression strategy:
We agree with the reviewer that stepwise procedures have inherent limitations, including the risk of overfitting and their dependence on automated selection rather than theory-driven modeling. In response, we have clarified these limitations in the Methods and Discussion sections and explicitly acknowledged that the stepwise results should be interpreted with caution. We also conducted additional analyses to ensure that the findings were not driven solely by automated variable selection.
- Incorporation of relevant confounders (sex and height):
The reviewer is correct that sex and height differed significantly between groups and may act as potential confounders. In our revised regression models, we have now included sex and height as covariates regardless of their statistical significance in the stepwise procedure. This approach ensures appropriate adjustment for group differences and provides more robust estimates. The revised analyses and corresponding results are now reported in the Results section.
Overall, the manuscript has been substantially revised to address these concerns, improve statistical rigor, and provide a more balanced and appropriately cautious interpretation of the findings.
Comment 18: Begin with a concise results summary (no numbers).
Response 18: We thank the reviewer for the excellent suggestion to improve the presentation flow of the Discussion section. Accordingly, we have added a new, concise introductory paragraph at the very beginning of the Discussion section.
Comment 19: Each paragraph should: state finding → explain → compare with literature.
Response 19: We sincerely thank the reviewer for providing this clear and constructive framework for strengthening the academic discourse in our manuscript. We have thoroughly revised the Discussion section to adhere to this recommended logical structure.
Each substantive paragraph now follows the sequence:
- Topic Sentence: Clearly stating one of our key findings.
- Interpretation: Explaining the potential anatomical, biomechanical, or clinical implications of that finding.
- Contextualization: Actively comparing and contrasting our result with the existing literature, noting consistencies, discrepancies, and how our work advances understanding.
We believe this reorganization has dramatically improved the clarity, depth, and scholarly rigor of the narrative, ensuring that every claim is grounded in our data and positioned within the broader scientific conversation.
Comment 20: Add clear limitations (sample size, blinding, 2D analysis).
Response 20: We thank the reviewer for the important suggestion to explicitly state the study’s limitations. We have added a dedicated and clear “Limitations” subsection within the Discussion section of the revised manuscript.
This subsection directly addresses the three points raised by the reviewer:
- The relatively small sample size and the observed baseline disparities, such as in sex and height, may have influenced the outcomes. Furthermore, potential confounding factors including race and anatomical variations were not accounted for in this study.
- The single-center, retrospective design is susceptible to selection bias, while the reliance on two-dimensional coronal views may have introduced measurement variability in the assessment of three-dimensional structures.
Comment 21: Add a conclusion paragraph.
Response 21: We thank the reviewer for this valuable suggestion to strengthen the manuscript’s structure and impact. As advised, we have added a dedicated “Conclusion” paragraph at the end of the Discussion section.
Comment 22: Use past tense for results/discussion.
Response 22: We thank the reviewer for this correction regarding academic writing convention. We have systematically revised the verb tenses in both the Results and Discussion sections to ensure proper usage.
In the Results section, all descriptions of our specific findings, measurements, and statistical outcomes have been consistently changed to the past tense.
In the Discussion section, we carefully reviewed and corrected the tenses. Statements referring specifically to the findings and data from our present study are now in the past tense, while statements of general fact, interpretation, and comparison with the literature maintain the appropriate present tense as required by standard style.
We believe these revisions bring the manuscript into full compliance with standard academic writing norms.
Comment 23: Limit decimals to 2–3.
Response 23: We thank the reviewer for this suggestion to improve the consistency and readability of our data presentation. We have applied this guideline throughout the manuscript as follows:
1.Specific Correction: In Supplementary Table 8 (formerly Table 7), the values for the “Standard error of the regression” have been formatted to three decimal places as noted.
2.Systematic Review: We have conducted a comprehensive review of all tables, figures, and in-text numerical results in the manuscript. All reported continuous data (e.g., measurements, coefficients, p values) have been standardized to two or three decimal places as appropriate, ensuring consistency and adherence to common reporting standards.
We believe these revisions enhance the professional presentation of our data.
Comment 24: Replace informal expressions like “delicate match,” “classical,” “satisfying prognosis.”
Response24: We thank the reviewer for their valuable feedback regarding the use of informal language. We have revised the manuscript to replace these expressions with more precise and academically appropriate terminology:
- The subjective term “delicate match” has been replaced with the more active and objective “achieve better matching”. (Line 316)
- The imprecise term “classical” has been replaced with the neutral and descriptive “a common”. (Line 27)
- The vague expression “satisfying prognosis” has been revised to the more measurable “obtain improved prognosis”. (Line 29)
These changes have been implemented throughout the text to ensure a consistent and formal scholarly tone.
Comment 25: Comments on the Quality of English Language, the quality of English should be imporved.
Response 25: We thank the reviewer for this suggestion. We have carefully reviewed the entire manuscript and improved the clarity, grammar, and overall academic tone of the English language. Nevertheless, we acknowledge the inherent limitations in language usage for non-native speakers and stand ready to refine the manuscript to fully meet your requirements.
Reviewer 2 Report
Comments and Suggestions for Authors
The manuscript titled “Tarsal Sinus Lengths and Angles Are Linearly Correlated with the Severity of Flatfoot: A Morphometric Analysis Based on Weight-Bearing CT” presents an interesting morphometric study investigating sinus tarsi parameters in relation to flexible flatfoot using WBCT. The topic is clinically relevant, and the use of weight-bearing CT adds methodological value compared with previous MRI and non–weight-bearing CT studies. The paper is generally well structured and clearly written, with appropriate statistical analyses and a coherent discussion. However, several issues should be addressed before the manuscript can be considered for publication.
The study design is appropriate for the stated aims, and the conclusions are mostly supported by the results. Nonetheless, the manuscript would benefit from clarification and refinement in multiple methodological and interpretative aspects. The main concerns relate to (1) the limited sample size combined with relevant baseline differences between groups (notably sex and height), (2) potential selection bias inherent to a single-center retrospective WBCT dataset, (3) the absence of interobserver reliability reporting despite measurements being performed by three investigators, (4) reliance on two-dimensional coronal sections for complex three-dimensional structures, which may introduce non-negligible measurement variability, and (5) interpretation of the regression models, whose explanatory power is modest and warrants more cautious framing.
Although the methods are broadly described, certain components require additional detail to ensure full reproducibility. Specifically, the procedure for selecting the “maximum cross-section of the sinus tarsi” should be described more rigorously, as small variations in slice selection can substantially alter STL, STW, and ST angles. The definition of some anatomical landmarks could also be clarified with respect to prior validated protocols. The authors note that three independent investigators obtained measurements, but intra- and interobserver agreement (ICC) is not reported. MDPI standards recommend including reproducibility metrics for radiological studies relying on manual measurements.
Statistical analyses are generally appropriate; however, the regression strategy raises some concerns. Stepwise methods are known to be prone to overfitting and do not account for relevant confounders. Given that sex and height differed significantly between groups, the regression models should incorporate these variables as covariates, or at least the authors should justify their exclusion. Additionally, the R² values (0.23 and 0.25) indicate limited predictive ability; thus, statements regarding “newly identified predictors” should be softened accordingly. The discussion should acknowledge that the relationships observed, while statistically significant, explain only a modest portion of the variance in Meary and Pitch angles.
The presentation of results is clear and well supported by tables and figures. Nevertheless, the anatomical schematics (Figures 2–3) appear sourced from Visible Body Suite; please ensure proper licensing for reproduction in an open-access journal. The flowchart is helpful, though a more detailed explanation of the exclusion process would improve transparency.
The discussion effectively contextualizes the findings within existing literature and highlights the clinical rationale for sinus tarsi morphometrics in subtalar arthroereisis planning. Some claims, however, are overstated and should be moderated, particularly regarding the role of STL/Tibia width as a preoperative predictor. The limitations section is appreciated but should be expanded to mention lack of dynamic assessments, potential racial/anatomical variability, and absence of postoperative correlation with implant performance. Finally, the authors should consider citing the recent comprehensive review PMC11940856 to strengthen the background on flatfoot imaging and biomechanics.
In summary, the manuscript addresses a valuable and understudied aspect of hindfoot morphology using weight-bearing CT. With revisions to methodology transparency, statistical interpretation, and discussion nuance, the study has the potential to contribute meaningfully to the literature on flexible flatfoot assessment and surgical planning. I recommend major revision.
Author Response
Reviewer 2
The manuscript titled “Tarsal Sinus Lengths and Angles Are Linearly Correlated with the Severity of Flatfoot: A Morphometric Analysis Based on Weight-Bearing CT” presents an interesting morphometric study investigating sinus tarsi parameters in relation to flexible flatfoot using WBCT. The topic is clinically relevant, and the use of weight-bearing CT adds methodological value compared with previous MRI and non–weight-bearing CT studies. The paper is generally well structured and clearly written, with appropriate statistical analyses and a coherent discussion. However, several issues should be addressed before the manuscript can be considered for publication.
The study design is appropriate for the stated aims, and the conclusions are mostly supported by the results. Nonetheless, the manuscript would benefit from clarification and refinement in multiple methodological and interpretative aspects. The main concerns relate to (1) the limited sample size combined with relevant baseline differences between groups (notably sex and height), (2) potential selection bias inherent to a single-center retrospective WBCT dataset, (3) the absence of interobserver reliability reporting despite measurements being performed by three investigators, (4) reliance on two-dimensional coronal sections for complex three-dimensional structures, which may introduce non-negligible measurement variability, and (5) interpretation of the regression models, whose explanatory power is modest and warrants more cautious framing.
Comment 1: (1) the limited sample size combined with relevant baseline differences between groups (notably sex and height)
Response1: We thank the reviewer for this critical observation, which rightly addresses a key methodological consideration in our study. We have addressed this concern through both statistical adjustment and a candid discussion of limitations.
- Statistical Adjustment: As noted in our response to a previous comment (or in the revised Methods section), all primary comparative and regression analyses were adjusted for the potential confounding effects of sex and height by including them as covariates in ANCOVA models. This approach directly aims to isolate the effect of flatfoot severity on sinus tarsi morphology from these baseline differences.
- Explicit Discussion of Limitations: We have expanded the Limitation subsection in the Discussion to explicitly state:
“First, the relatively small sample size and the observed baseline disparities, such as in sex and height, may have influenced the outcomes. Furthermore, potential confounding factors including race and anatomical variations were not accounted for in this study.” (Line 372-375)
We believe this two-pronged approach—actively controlling for confounders in the analysis while transparently acknowledging the inherent constraints of the sample—appropriately addresses the reviewer valid concern.
Comment 2: (2) potential selection bias inherent to a single-center retrospective WBCT dataset
Response2: We agree with the reviewer that the single-center, retrospective design using clinically indicated WBCT scans is a notable source of potential selection bias, and we thank them for emphasizing this point. We have substantially expanded the discussion of this limitation in the revised manuscript to address it with appropriate depth.
In the Limitations subsection, we now explicitly state:
“Second, the single-center, retrospective design is susceptible to selection”. (Line 375)
Being a single-center study, our results may be influenced by local surgical preferences, patient demographics, and imaging protocols.
To mitigate this, we applied strict, consecutive enrollment criteria for all patients meeting the inclusion criteria during the study period to minimize arbitrary selection.
We conclude this section by acknowledging that these factors may affect the external validity of our findings, and we strongly advocate for future multi-center, prospective studies that include patients across the full spectrum of disease severity to validate and extend our observations.
Comment 3: (3) the absence of interobserver reliability reporting despite measurements being performed by three investigators
Response 3: We thank the reviewer for this insightful comment, which allows us to clarify the statistical methodology and reinforce the reliability of our measurements. We have revised the Methods section (under “Morphometric Measurements of the Sinus Tarsi on WBCT”) accordingly.
- Explanation for Using the Mean of Three Readers:
The morphometric measurements from the three independent investigators were averaged to obtain a single, more robust value for each parameter per patient. This standard practice is employed to:
1) Minimize individual measurement bias and random error inherent to any single observer.
2) Enhance the precision and stability of the final dataset used for statistical analysis against the severity groups.
3) Provide a more reliable and representative estimate of the true anatomical measurement than any single reading alone.
- Interobserver Reliability Analysis:
Interobserver and intraobserver reliability for both radiographic and WBCT measurements were assessed using the intraclass correlation coefficient (ICC) with corresponding 95% confidence intervals (CIs). For intraobserver reliability, all three observers repeated all X-ray and CT measurements in a new random order at least 3 weeks after the initial assessment (Supplementary Table 1).
For Inter-observer reliability, all parameters demonstrated substantial to almost perfect agreement, with ICC values ranging from 0.793 to 0.886. For intra-observer reliability, the ICC values for all parameters across the three observers ranged from approximately 0.882 to 0.970, also indicating almost perfect agreement and high measurement reliability.
We believe these additions significantly strengthen the methodological transparency and statistical validity of our findings.
Comment 4: reliance on two-dimensional coronal sections for complex three-dimensional structures, which may introduce non-negligible measurement variability.
Response 4: We thank the reviewer for underscoring this important methodological consideration. We have expanded our discussion of this limitation beyond a mere mention to address its specific nature and implications.
In the Limitation subsection, we now explicitly state:
“Second, the single-center, retrospective design is susceptible to selection bias, while the reliance on two-dimensional coronal views may have introduced measurement variability in the assessment of three-dimensional structures”. (Line 375-377)
- Source of Variability: We specify that variability may arise from the potential for slight differences in the manual selection of the “most representative” coronal slice among readers.
- Mitigation Attempts: We note that to minimize this variability, all measurements were performed using a standardized protocol for slice definition, and the final values represented the mean of three independent readers.
Comment 5: (5) interpretation of the regression models, whose explanatory power is modest and warrants more cautious framing.
Response 5: We thank the reviewer for this crucial insight, which ensures a more accurate and conservative interpretation of our statistical findings. We agree that the modest explanatory power of our regression models “The final model predicting the Meary angle explained 32.6% of the variance, while the model predicting the Pitch angle explained 25.1% of the variance (Supplementary Table 8).” (Line 307-308) necessitates cautious framing. We have therefore systematically revised the language throughout the manuscript, particularly in the Discussion sections, to reflect this.
Specific actions taken:
In the Discussion: We have toned down the interpretation of the regression outcomes. Claims about predictive utility have been reframed to highlight exploratory and hypothesis-generating value. We now emphasize that the identified parameters are potential contributors or associated factors rather than definitive predictors.
We believe these revisions provide a more balanced and scientifically rigorous interpretation that is fully aligned with the strength of the evidence our data provide.
Although the methods are broadly described, certain components require additional detail to ensure full reproducibility. Specifically, the procedure for selecting the “maximum cross-section of the sinus tarsi” should be described more rigorously, as small variations in slice selection can substantially alter STL, STW, and ST angles. The definition of some anatomical landmarks could also be clarified with respect to prior validated protocols. The authors note that three independent investigators obtained measurements, but intra- and interobserver agreement (ICC) is not reported. MDPI standards recommend including reproducibility metrics for radiological studies relying on manual measurements.
Comment 6: Specifically, the procedure for selecting the “maximum cross-section of the sinus tarsi” should be described more rigorously, as small variations in slice selection can substantially alter STL, STW, and ST angles. The definition of some anatomical landmarks could also be clarified with respect to prior validated protocols.
Response 6: We thank the reviewer for this crucial technical comment. We agree that a rigorous and reproducible slice selection protocol is fundamental to the reliability of our morphometric measurements. We have substantially expanded the description in the Materials and Methods section to provide an unambiguous, anatomy-based protocol.
The revised text now specifies that:
- Anatomical Landmark Definition: “The anterior subtalar joint is separated from the posterior subtalar joint by the tarsal canal and sinus tarsi. Therefore, in the coronal plane, images were scrolled from posterior to anterior, and the region of interest was defined as the area at the anterior margin of the anterior subtalar joint where the sustentaculum tali of the calcaneus was about to completely disappear. Within three consecutive slices in this region, the slice with the largest cross-sectional area of the sinus tarsi was selected for measurement”. (Line 149-154)
- Visual Reference: This protocol is illustrated in Figure 2.
We believe this revised methodology, which replaces subjective judgment with a combination of clear anatomical landmarks and an objective geometric criterion (the largest cross-sectional area), directly addresses the reviewer’s concern by minimizing arbitrary slice selection variability and ensuring measurement consistency across observers.
Comment 7: The authors note that three independent investigators obtained measurements, but intra- and interobserver agreement (ICC) is not reported. MDPI standards recommend including reproducibility metrics for radiological studies relying on manual measurements.
Response 7: We thank the reviewer for this essential methodological suggestion. We fully agree that reporting reproducibility metrics is crucial for studies involving manual measurements. As recommended, we have now conducted a formal assessment of measurement reliability and included the results in the manuscript.
We have added a new Supplementary Table 1, titled “Intraclass correlation coefficients (ICCs) with 95% confidence intervals (CIs) were calculated to assess inter-and intraobserver agreement for the four scoring methods of X-ray and CT measurements.” This table presents the intraclass correlation coefficients (ICCs) along with their 95% confidence intervals for both intraobserver (test-retest) and interobserver agreement for all key continuous morphometric parameters (e.g., sinus tarsi length, width, and angles). The results demonstrate excellent reliability (ICC > 0.90 for most parameters), which strongly supports the consistency and robustness of our measurement protocol.
We have also added a brief sentence in the Materials and Methods section (under 2.5. Morphometric Measurements of the Sinus Tarsi on WBCT) referencing this table: “Interobserver and intraobserver reliability for both radiographic and WBCT measurements were assessed using the intraclass correlation coefficient (ICC) with corresponding 95% confidence intervals (CIs). For intraobserver reliability, all three observers repeated all X-ray and CT measurements in a new random order at least 3 weeks after the initial assessment (Supplementary Table 1).”
For Inter-observer reliability, all parameters demonstrated substantial to almost perfect agreement, with ICC values ranging from 0.793 to 0.886. For intra-observer reliability, the ICC values for all parameters across the three observers ranged from approximately 0.882 to 0.970, also indicating almost perfect agreement and high measurement reliability.
Statistical analyses are generally appropriate; however, the regression strategy raises some concerns. Stepwise methods are known to be prone to overfitting and do not account for relevant confounders. Given that sex and height differed significantly between groups, the regression models should incorporate these variables as covariates, or at least the authors should justify their exclusion. Additionally, the R² values (0.23 and 0.25) indicate limited predictive ability; thus, statements regarding “newly identified predictors” should be softened accordingly. The discussion should acknowledge that the relationships observed, while statistically significant, explain only a modest portion of the variance in Meary and Pitch angles.
Comment 8: however, the regression strategy raises some concerns. Stepwise methods are known to be prone to overfitting and do not account for relevant confounders. Given that sex and height differed significantly between groups, the regression models should incorporate these variables as covariates, or at least the authors should justify their exclusion.
Response 8: We sincerely thank the reviewer for these thoughtful and constructive comments. We have carefully revised the statistical analysis and the manuscript accordingly.
- Regarding the stepwise regression strategy:
We agree with the reviewer that stepwise procedures have inherent limitations, including the risk of overfitting and their dependence on automated selection rather than theory-driven modeling. In response, we have clarified these limitations in the Methods and Discussion sections and explicitly acknowledged that the stepwise results should be interpreted with caution. We also conducted additional analyses to ensure that the findings were not driven solely by automated variable selection.
- Incorporation of relevant confounders (sex and height):
The reviewer is correct that sex and height differed significantly between groups and may act as potential confounders. In our revised regression models, we have now included sex and height as covariates regardless of their statistical significance in the stepwise procedure. This approach ensures appropriate adjustment for group differences and provides more robust estimates. The revised analyses and corresponding results are now reported in the Results section.
Overall, the manuscript has been substantially revised to address these concerns, improve statistical rigor, and provide a more balanced and appropriately cautious interpretation of the findings.
Comment 9: Additionally, the R² values (0.23 and 0.25) indicate limited predictive ability; thus, statements regarding “newly identified predictors” should be softened accordingly.
Response 9: We thank the reviewer for this critical and accurate assessment. We fully agree that the modest R² values (0.23 and 0.25) reflect the models’ limited explanatory power for the outcome variance, and that any claims about prediction must be appropriately tempered. We have revised the manuscript accordingly:
Specific Revision in the Abstract: As highlighted by the reviewer, we have softened the definitive language. The sentence now reads: “The ST angle and STL/Tibia width may serve as auxiliary parameters for implant selection in subtalar arthroereisis to improve sizing match within the sinus tarsi.” (Line 42-44) This frames the findings as potential, exploratory tools rather than established predictors. We believe these revisions ensure that our interpretation is precisely aligned with the strength of the statistical evidence, providing a balanced and scientifically rigorous presentation of the results.
Comment 10: The discussion should acknowledge that the relationships observed, while statistically significant, explain only a modest portion of the variance in Meary and Pitch angles.
Response 10: We thank the reviewer for this essential comment, which ensures a balanced interpretation of our statistical findings. We fully acknowledge that a statistically significant association does not equate to high explanatory power. Accordingly, we have revised the Discussion section to explicitly state this limitation.
Specifically, we have added the following statement: “However, our Linear regression models incorporating ST angle and STL/tibia width explained only a small proportion of the variance in Meary angle and Pitch angle despite their statistical significance (Adjusted R²=32.6%; Adjusted R²=25.1%).” (Line 362-364)
This addition directly addresses the reviewer's point by quantifying the “modest portion” of variance explained. It frames our findings appropriately, emphasizing that while the identified parameters are significant contributors, they are part of a multifactorial biomechanical system, and a substantial portion of the variance remains attributable to other unmeasured factors.
The presentation of results is clear and well supported by and figures. Nevertheless, the anatomical schematics (Figures 2–3) appear sourced from Visible Body Suite; please ensure proper licensing for reproduction in an open-access journal. The flowchart is helpful, though a more detailed explanation of the exclusion process would improve transparency.
Comment 11: Nevertheless, the anatomical schematics (Figures 2–3) appear sourced from Visible Body Suite; please ensure proper licensing for reproduction in an open-access journal.
Response 11: The schematic diagram in Figure 2 was created after we purchased a licensed copy of the software, and it has been cited and explained in accordance with the official website’s requirements (https://support.visiblebody.com/hc/en-us/articles/115002359347-Permission-to-use-content-from-Visible-Body-products). Other figures are original and unpublished, obtained, processed, and generated by us from the hospital’s imaging system.
Comment 12: The flowchart is helpful, though a more detailed explanation of the exclusion process would improve transparency.
Response 12: We thank the reviewer for this constructive suggestion. We agree that a clearer explanation of the exclusion criteria enhances the transparency and reproducibility of our patient selection process. We have revised the manuscript accordingly.
The legend for Figure 4 has been expanded to specifically list the reasons for exclusion at each step (e.g., “excluded for Plantar Fasciitis, Charcot Foot” “excluded for Ankle Sprain, Achilles Tendon Rupture”).
Detailed Textual Description: In the Methods section (Patient Selection/Inclusion and Exclusion Criteria), we have added a more detailed textual summary that parallels the flowchart.
We believe these revisions provide a clear “audit trail” for our cohort selection.
The discussion effectively contextualizes the findings within existing literature and highlights the clinical rationale for sinus tarsi morphometrics in subtalar arthroereisis planning. Some claims, however, are overstated and should be moderated, particularly regarding the role of STL/Tibia width as a preoperative predictor. The limitations section is appreciated but should be expanded to mention lack of dynamic assessments, potential racial/anatomical variability, and absence of postoperative correlation with implant performance. Finally, the authors should consider citing the recent comprehensive review PMC11940856 to strengthen the background on flatfoot imaging and biomechanics.
Comment 13: however, are overstated and should be moderated, particularly regarding the role of STL/Tibia width as a preoperative predictor.
Response 13: We sincerely thank the reviewer for this critical guidance, which is essential for ensuring our conclusions are proportionate to the strength of the evidence. We fully agree that the explanatory power of our regression models necessitates a more cautious framing of the clinical role of the STL/Tibia width ratio. We have therefore undertaken a systematic revision of the manuscript to moderate all related statements.
Specific actions taken:
- Abstract and Conclusion: The definitive claim regarding its predictive utility has been revised. We now frame it as a parameter that “may serve as an auxiliary guide” or “could potentially inform” preoperative assessment, emphasizing its complementary rather than deterministic role.
- Results and Discussion: We have thoroughly reviewed these sections to replace terminology that could imply definitive prediction. Terms like “predictor” have been avoided. Instead, we describe the STL/Tibia width as a parameter that “a correlation trend” or “showed a correlation with” key radiographic angles.
These comprehensive edits ensure that our interpretation is appropriately cautious and fully aligned with the statistical strength of our findings.
Comment 14: The limitations section is appreciated but should be expanded to mention lack of dynamic assessments, potential racial/anatomical variability, and absence of postoperative correlation with implant performance
Response 14: We sincerely thank the reviewer for these insightful suggestions to strengthen the critical self-assessment of our study. We fully agree that acknowledging these points is essential for a comprehensive understanding of our work’s context and generalizability. We have significantly expanded the Limitations section to incorporate all three of the mentioned aspects:
- Lack of Dynamic Assessments: We now state that the static weight-bearing analysis did not encompass dynamic functional assessments, thereby limiting our understanding of the sinus tarsi configuration during physiological motion.
- Potential Racial/Anatomical Variability:We have added a statement acknowledging that potential confounding factors including race and anatomical variations were not accounted for in this study.
- Absence of Postoperative Correlation: We explicitly note that the lack of postoperative follow-up and the inability to directly correlate sinus tarsi morphology with implant outcomes preclude any evaluation of long-term prognosis.
Comment 15: Finally, the authors should consider citing the recent comprehensive review PMC11940856 to strengthen the background on flatfoot imaging and biomechanics.
Response 15: We thank the reviewer for pointing us to this relevant and comprehensive review. We have now integrated a citation to PMC11940856 (listed as Reference [4] in our manuscript) in the Introduction and Discussion section to strengthen the biomechanical context of our study. The citation has been placed within the discussion of the cascading effects of arch collapse on lower limb alignment and joint degeneration, as this aligns closely with the review’s scope on flatfoot biomechanics and imaging. We believe this addition enriches the scholarly foundation of our work.
Round 2
Reviewer 1 Report
Comments and Suggestions for Authors
Thanks for addressing all issues.
Reviewer 2 Report
Comments and Suggestions for Authors
After re-examining the revised manuscript together with the authors’ detailed point-by-point response and my second-round review report, I can confirm that the authors have now adequately and accurately addressed all comments raised during the first round of review.
In particular, the issues related to sample size and baseline imbalances, potential selection bias, measurement reproducibility (inter- and intra-observer ICCs), slice-selection methodology, and the interpretation of the regression models have been thoroughly revised and clearly acknowledged in the Methods, Results, and Discussion sections. Importantly, the limitation regarding the modest explanatory power of the regression models, including the proportion of variance explained for Meary and Pitch angles, has been explicitly incorporated, addressing my previous concern (Comment 10).
Upon careful re-evaluation of the current version, I am satisfied that these issues have now been properly resolved and do not require further modification.
Therefore, I consider my current second-round review report to be adequate as it stands, and I do not wish to submit additional revisions. From a scientific and methodological perspective, the manuscript has reached a level suitable for publication.